# Cell Wall Components and Extensibility Regulate Root Growth in *Suaeda salsa* and *Spinacia oleracea* under Salinity

**DOI:** 10.3390/plants11070900

**Published:** 2022-03-28

**Authors:** Jia Liu, Yang Shao, Xiaohui Feng, Victoria Otie, Asana Matsuura, Muhammad Irshad, Yuanrun Zheng, Ping An

**Affiliations:** 1Arid Land Research Center, Tottori University, 1390, Hamasaka, Tottori 680-0001, Japan; liujia.alrc@gmail.com; 2College of Plant Science & Technology, Huazhong Agricultural University, Wuhan 430070, China; barnett@mail.hzau.edu.cn; 3Key Laboratory of Water Cycle and Related Land Surface Processes, Institute of Geographic Sciences and Natural Resources Research, Chinese Academy of Sciences, 11 A Datun Road, Beijing 100101, China; fxhcaf@163.com; 4Department of Soil Science, Faculty of Agriculture, Forestry and Wildlife Resources Management, University of Calabar, Calabar P.M.B. 1115, Nigeria; victoriaotie@unical.edu.ng; 5Faculty of Agriculture, Shinshu University, 8304, Minamiminowa-Village, Kamiina-County, Nagano 399-4598, Japan; asana@shinshu-u.ac.jp; 6Department of Environmental Sciences, Abbottabad Campus, COMSATS University Islamabad (CUI), Abbottabad 22060, Pakistan; mirshad@ciit.net.pk; 7Laboratory of Resource Plants, West China Subalpine Botanical Garden, Institute of Botany, Chinese Academy of Sciences, Beijing 100093, China; zhengyr@ibcas.ac.cn

**Keywords:** cell wall integrity, glycophyte, halophyte, *Suaeda salsa*, *Spinacia oleracea*, salinity stress

## Abstract

Understanding the role of root cell walls in the mechanism of plant tolerance to salinity requires elucidation of the changes caused by salinity in the interactions between the mechanical properties of the cell walls and root growth, and between the chemical composition of the cell walls and root growth. Here, we investigated cell wall composition and extensibility of roots by growing a halophyte (*Suaeda salsa*) and a glycophyte (*Spinacia oleracea*) species under an NaCl concentration gradient. Root growth was inhibited by increased salinity in both species. However, root growth was more strongly reduced in *S. oleracea* than in *S. salsa*. Salinity reduced cell wall extensibility in *S. oleracea* significantly, whereas treatment with up to 200 mM NaCl increased it in *S. salsa*. Meanwhile, *S. salsa* root cell walls exhibited relatively high cell wall stiffness under 300 mM NaCl treatment, which resist wall deformation under such stress conditions. There was no decrease in pectin content with salinity treatment in the cell walls of the elongation zone of *S. salsa* roots. Conversely, a decrease in pectin content was noted with increasing salinity in *S. oleracea*, which might be due to Na^+^ accumulation. Cellulose content and uronic acid proportions in pectin increased with salinity in both species. Our results suggest that (1) cell wall pectin plays important roles in cell wall extension in both species under salinity, and that the salt tolerance of glycophyte *S. oleracea* is affected by the pectin; (2) cellulose limits root elongation under saline conditions in both species, but in halophytes, a high cell wall content and the proportion of cellulose in cell walls may be a salt tolerance mechanism that protects the stability of cell structure under salt stress; and (3) the role of the cell wall in root growth under salinity is more prominent in the glycophyte than in the halophyte.

## 1. Introduction

Soil salinity is an environmental factor that severely limits global agricultural productivity. An understanding of the mechanisms underlying salt tolerance would contribute to global crop production [1]. The growth response of halophytes and glycophytes to salinity can differ significantly. Halophytes show high salt tolerance and can complete their life cycle under salt concentrations that reach 600 mM NaCl [2]. In contrast, glycophyte root growth is severely hindered when soil salinity exceeds 70 mM NaCl [3]. The study of the physiological responses of glycophytes and halophytes to salinity is useful for understanding the mechanisms underlying plant responses to salinity. Recent studies on the mechanism of salt tolerance in plants have focused on the symplast pathway [4], while studies on the apoplast pathway are relatively limited.

The cell wall matrix is composed of three types of polysaccharides: pectin, hemicellulose, and cellulose. The structure, function, composition, and linkage of cell wall components have been reviewed by Cosgrove [5,6]. However, less is known about the salt-tolerance-related functions of root cell walls, which directly interact with salts present in the soil solution and in the plant. Saline treatment changes the chemical composition of glycophyte root cell walls. For example, sodium ions can displace Ca^2+^ from their binding sites on the cell wall, which can reduce pectin-crosslinking and cell wall integrity [7,8,9,10]. Furthermore, the cellulose content can decrease when uronic acid increases in cotton grown in saline conditions [11]. Conversely, although pectin content decreases in soybean apical root tips in saline conditions, cellulose content increases [12]. In *Artemisia annua*, the monosaccharide composition of pectin and hemicellulose was altered in response to salt stress [13]. Similarly, salinity increased the concentration of carboxylated polysaccharides in *Oryza sativa* [14]. The cell wall of stele cells in citrus plants can also influence root Na^+^ transport by acting as an Na^+^ trap [15]. Furthermore, the Na^+^ concentration in the cortical cell walls of a salt-sensitive barley cultivar was approximately twice that observed in the cortical cell walls of a salt-tolerant cultivar, indicating that cell wall composition can influence Na^+^ transport [16].

Previous studies have demonstrated that the mechanical and chemical characteristics of the root cell wall are associated with root growth under abiotic stress conditions [17,18]. The cell wall’s extensibility is an important trait in the regulation of cell elongation under different abiotic stress conditions [19,20]. Plastic deformability in the root elongation zone of soybean roots is reduced by salt stress [21], and the extensibility of root cell walls is inhibited by water deficit in maize and damask rose [22,23]. Furthermore, the changes in cell wall extensibility are associated with root elongation in sorghum plants treated with silicon [24]; SiO_2_ treatment increases cell wall pectin content and physical strength [25]. In contrast, cell wall extensibility and root elongation are inhibited in maize by the binding of Al^3+^ to the cell wall [26] and cause the accumulation of cell wall polysaccharides [27]. Similarly, lead (Pb) accumulates mainly in the roots, where it has a high affinity for galacturonic acid [28], which may change pectin structure and decrease cell wall extensibility. Finally, Pb has been found to reduce cell wall extensibility by influencing the synthesis of cell wall polysaccharides [29].

Despite this wealth of information, there are no reports of changes in the interactions between the mechanical properties of root cell walls and root growth or between the chemical composition of root cell walls and root growth under saline conditions. Therefore, we investigated cell wall composition, extensibility, and viscosity in the root elongation zone of young seedlings of a halophyte (*S. salsa* L.) and a glycophyte (*S. oleracea* L.), which both belong to the Amaranthaceae family (while they were formerly grouped to Chenopodiaceae). These plants were grown under an NaCl concentration gradient so we could compare the response of each species to salinity to understand the mechanisms of plant salt tolerance better.

## 2. Material and Methods

### 2.1. Plant Materials

Seeds of *S. salsa* were collected in the coastal saline region of Bohai Bay, Hebei Province, China. The seeds of *S. oleracea* ‘Akinokagayaku’ were bought in a local seed market. Although they belong to the same family, *S. oleracea* is sensitive to salinity stress, indicated by its relatively low ability to counteract salinity by salt exclusion [30]. *S. salsa* is relatively more tolerant to salt stress than *S. oleracea* due to its ability to accumulate large concentrations of salts in its vacuoles [31].

### 2.2. Seedling Growth and Salt Treatment

Seeds of *S. salsa* were surface sterilized with 5% NaClO for 10 min and then thoroughly rinsed with water. Seed germination and seedling growth were conducted in growth chambers (MLR-350HT; Sanyo, Osaka, Japan). Temperature and relative humidity of the chambers were 20 °C and 70%, respectively. Approximately 15–20 seeds were aligned on filter paper and placed in a zip-lock plastic bag. The filter paper was rinsed every day with 1/12 diluted Hoagland solution (pH 6.5) during the two-day germination period for *S. salsa*, and over a four-day period for *S. oleracea*. The seeds were germinated in the dark. After germination, the roots of the seedlings were rinsed every 2 days with 1/12 diluted Hoagland solution mixed with 0, 100, 200, and 300 mM NaCl. The seedlings were subjected to the salinity treatments for 8 days. Light conditions were set to a 12 h/12 h light/dark cycle. Each salt concentration treatment group included 24 filter paper germination sheets. After germination was completed, five filter paper sheets were randomly selected at 2 days intervals from each treatment to measure the root length of each seedling. As we found in the preliminary experiment, 1/12 diluted Hoagland solution did not cause nutrient deficiency, and exposure to high salt concentration did not cause growth arrest through osmotic shock. Therefore, we chose 1/12 diluted Hoagland solution to reveal the salt tolerance of these species and their differences when grown in graduated low nutrient conditions.

### 2.3. Mechanical Parameters of the Root Cell Wall

Root samples from separately cultured seedlings were excised from the apical zone (extending 15 mm behind the root tip) and immediately transferred to boiling methanol in a water bath heated to approximately 80 °C. Samples were incubated in this solution for 5 min. Methanol-killed root segments were rehydrated with 1/12 diluted Hoagland solution (pH 6.5), and cell wall elastic moduli and viscosity coefficients were measured using a creep meter (RE2-33005C-1,2; Yamaden, Tokyo, Japan). This 2–5 mm root segment from the apical zone was fixed between the two clamps of the creep meter, and a tensile force along the direction of root elongation of 0.025 to 0.1 N was applied to root samples depending on the measured diameter. We dropped a drip of water at the basis of root section before the extension measurement and kept a humid environment around the creepmeter using a humidifier. The extension process was electronically recorded at intervals of 0.5 s for 300 s. The load was then released, and root shrinkage was recorded for 300 s. The difference in length between the maximum extension after 300 s and the final length after 600 s was reported as a reversible extension (i.e., viscoelastic extension), while the difference between final and original lengths (3 mm) was reported as a plastic extension. The elastic modulus (E_0_) and viscosity coefficient (η_N_) [32] were determined using the creep meter software supplied. We measured 15 root segments from each NaCl treatment.

### 2.4. Isolation of Root Cell Wall Polysaccharides

Seedlings were prepared as described above. Seedling roots were thoroughly washed and cut into two 5 mm segments. The 0–5 mm segment measured from behind the root cap was considered the elongation zone [33], and the 5–10 mm segment measured from behind the root cap were sampled. Approximately 96–120 segments from six filter paper sheets were collected as one replicate, and each treatment has four replicates.

Cell wall polysaccharides were isolated, as previously described [34]. Briefly, the root segments were homogenized for 30 s in a mixture of ice-cold Tris-HCl buffer (pH 7.4) and Tris buffer-saturated phenol using a bead crusher (Model μT-12; TAITEC Co., Ltd., Tokyo, Japan). The homogenate was centrifuged (Kubota 6200; KUBOTA Corporation Co., Ltd., Tokyo, Japan) for 10 min at 4 °C, and the supernatant discarded. The pellet containing the cell walls was further purified by sequential incubation and centrifugation in ethanol (thrice), acetone (twice), a mixture of methanol and chloroform (1:1, *v*/*v*) (thrice), and again in acetone and ethanol. Cell walls were treated with 0.2 mg mL^−1^ pronase in 0.05 mM phosphate buffer (pH 7.0) containing 5% ethanol for 16 h at 30 °C. After centrifugation, the supernatant was discarded.

Cell wall polysaccharides were extracted as described by An et al. [12]. Briefly, the pectin fractions were extracted five times with CDTA at pH 6.5 at 20 °C. To extract the remaining polyuronides, cell walls were further extracted three times with CDTA at 100 °C (hot CDTA) for 1 h each. These CDTA extractions were designated as the pectin fraction. Hemicellulose I and II were sequentially extracted with 1 M and 4 M KOH, respectively. These extractions in KOH solutions were repeated three times for 4, 16, and 4 h each, respectively. Residual alkaline-insoluble precipitates were designated as the cellulose fraction, which was dissolved in a small amount of 72% (*v*/*v*) sulfuric acid for 1 h and diluted with distilled water to measure the amount of total sugars. The amounts of uronic acid and total sugars in each extract were measured using the *m*-hydroxydiphenyl colorimetric [35] and phenol-sulfuric acid [36] methods, respectively.

### 2.5. Statistical Analysis

All data were analyzed using the analysis of variance (ANOVA) and correlation; means were compared using Tukey’s Honestly Significant Difference test (*p* < 0.05). All statistical analyses were performed using SPSS software version 28.0 (SPSS, Inc., Chicago, IL, USA).

## 3. Results 

### 3.1. Root Growth

Figure 1 shows the root length and salt resistance coefficient of the two species when treated with different salt concentrations for 8 days after germination. Root growth of *S. oleracea* seedlings was inhibited by salinity and almost ceased in plants treated with 200 mM NaCl. Conversely, root growth was not inhibited in *S. salsa* plants treated with either 100 mM or 200 mM NaCl. Salinity had a significant negative effect on root growth with 300 mM NaCl in the halophytic species *S. salsa*. The inhibition was more pronounced in *S. oleracea* than in *S. salsa*.

### 3.2. Mechanical Properties of Root Cell Wall

Figure 2 shows E_0_ and η_N_ recorded for the root cell walls in the 2–5 mm root tip zone in *S. salsa* and *S. oleracea* seedlings growing under different NaCl concentrations. Lower values of E_0_ indicate higher cell wall elasticity, while greater values of η_N_ indicate higher cell wall viscosity [32]. E_0_ and η_N_ in *S. oleracea* root tips increased with increasing salinity, whereas 100 mM NaCl treatment reduced E_0_ but 300 mM NaCl treatment risen it in *S. salsa*. In all treatments, E_0_ was higher in *S. salsa* than in *S. oleracea*. η_N_ was higher in *S. salsa* than in *S. oleracea* under 0 and 300 mM NaCl treatments, whereas the opposite was observed in the 100 and 200 mM NaCl treatments.

The regression of root length on E_0_ revealed significant negative correlations for each species (Figure 3). The effect of E_0_ on root growth was considerably more pronounced in *S. oleracea* than in *S. salsa*. Similarly, the regression of root length on η_N_ also showed negative correlations for the two species, and again, the effect of η_N_ was more pronounced in *S. oleracea* than in *S. salsa* (Figure 3). In *S. oleracea*, the viscoelastic extension was reduced by salt treatments. In *S. salsa*, the plastic extension showed negative values, except in the 200 mM treatment. The negative value of plastic extension means that the root shrunk even shorter than the original length after extension. In contrast, this phenomenon was not found in *S. oleracea* (Figure 4).

### 3.3. Chemical Composition of the Root Cell Wall

Table 1 shows the effect of salt treatment on total sugar content in root cell walls. In the 0–5 mm segment behind the root tip, total sugar content was reduced at 100 mM NaCl treatment in both species. The same trend was observed in both 0–5 and 5–10 mm root segments; compared with 100 mM NaCl, 300 mM NaCl significantly increased total sugar content in both species. Moreover, sugar content was nearly two-fold greater in *S. salsa* than in *S. oleracea* across saline treatments. Furthermore, total sugar content was higher in the 5–10 mm root segment than in the 0–5 mm root segment.

Figure 5 and Figure 6 show the total sugar content in each cell wall fraction in *S. salsa* and *S. oleracea* under different NaCl concentrations. Pectin content decreased in the 0–5 mm region in *S. salsa* root tips at 100 and 300 mM NaCl. Hemicellulose I, hemicellulose II, and cellulose contents initially declined and then increased with increasing saline conditions in *S. salsa* root tips. In contrast, in *S. oleracea*, treatment with NaCl caused a significant decrease in pectin content, coupled with a significant increase in cellulose content at 300 mM NaCl. There was no change in hemicellulose I or hemicellulose II with NaCl treatment (Figure 5) in *S. oleracea*. In the 0–5 mm region of the root tips, we observed the same trend in the changes of hemicellulose and cellulose in *S. salsa*, with their content being much higher than those in *S. oleracea*. In the 5–10 mm root segments of *S. salsa*, pectin content decreased at 100 and 300 mM NaCl and cellulose content decreased at 100 mM NaCl (Figure 6). Hemicellulose I content decreased in *S. oleracea* at 100 and 200 mM NaCl. Pectin content in the 5–10 mm segments was lower than in the 0–5 mm root segments. Cellulose content in 5–10 mm root segments was higher than in the 0–5 mm root segments across treatments (Figure 5 and Figure 6). It is worth noting that, except for hemicellulose II, sugar contents of the cell wall fractions in *S. salsa* were higher at 0 mM than at 100 mM NaCl (Figure 6).

Uronic acid content in the different cell wall fractions is shown in Figure 5 and Figure 6. Uronic acid content within the pectin fraction decreased in the 0–5 mm root segments in *S. oleracea* under salinity but remained unchanged except for an increase at 200 mM NaCl in *S. salsa*. An increasing trend in uronic acid proportion in the hemicellulose I, hemicellulose II, and cellulose fractions was observed for both species in response to increasing salt concentration. Uronic acid content in each cell wall fraction generally increased in the 5–10 mm root-tip segments as salt concentration increased from 100 to 300 mM NaCl in both species. In both the 0–5 and 5–10 mm root-tip segments, uronic acid content was significantly higher in the cellulose of *S. salsa* than of *S. oleracea*.

Results of the correlation analysis are shown in Table 2 and Table 3. Hemicellulose I and hemicellulose II contents, including the uronic acid within them, were negatively correlated with root growth in *S. salsa*. In contrast, there was a positive correlation between pectin content and root growth in *S. oleracea*, while cellulose content was negatively correlated with root length in both species. Similarly, there was a positive correlation between the uronic acid content in pectin and root length in *S. oleracea*, but not in *S. salsa*. Furthermore, the uronic acid content in cellulose was negatively correlated to root length in both species. Lastly, pectin content in *S. oleracea* showed significant correlations with root length, cellulose content, and uronic acid content in pectin, hemicellulose II, and cellulose, whereas no such correlations were observed for *S. salsa*.

## 4. Discussion

A significant negative effect of salinity on root growth was evident even in *S. salsa*, indicating that these plants are highly sensitive to salinity during early growth. This finding substantiates the use of young seedlings as a suitable experimental material for studies on salt tolerance in plants. Furthermore, the pronounced inhibition in the root growth in *S. oleracea* grown under salt stress showed relatively greater sensitivity to salinity when compared with *S. salsa* (Figure 1 and Appendix A). Thus, these two related but contrasting species offer a unique opportunity to study the mechanisms underlying species-specific differences in salt-stress response.

Salt stress causes significant cell wall stiffening, which has a detrimental effect on root growth [37]. Thus, the changes in root growth may correlate with cell wall extensibility of the root elongation zone. Among the viscoelastic parameters measured in creep-extension, E_0_ most represented the overall root extensibility, with greater E_0_ values reported for stiffer cell walls, while greater η_N_ values indicate higher cell wall viscosity [32]. Thus, the negative correlation observed in this study between E_0_ and root growth in *S. salsa* and *S. oleracea* (Figure 3) indicated that cell extensibility in the root elongation zone is an important limiting factor of root growth in both halophyte and glycophyte species under saline conditions. Furthermore, the greater value for the regression slope of *S. oleracea* (3.45) than for *S. salsa* (0.21) clearly demonstrated that cell wall stiffness affects root growth more severely in salt-sensitive than in salt-tolerant species via changing extensibility (Figure 3). Higher cell wall extensibility was found to be favorable for root growth under saline conditions. Root cell wall extensibility in *S. salsa* seedlings changed minimally, even when treated with 200 mM NaCl, which would benefit root growth. In contrast, the increased values for E_0_ under salt treatment may have resulted in the reduced root growth in *S. oleracea* (Figure 1, Figure 2 and Figure 3). The similarity in the viscosity and elasticity of the root cell wall in response to saline conditions suggests that viscosity is also related to root growth (Figure 1, Figure 2 and Figure 3). The steeper slopes of the regressions between η_N_ and root growth in *S. oleracea* than in *S. salsa* indicated that η_N_ affected root growth in the salt-sensitive species more than in the salt-tolerant species grown in saline conditions (Figure 3). It also showed that extensibility and viscosity are distinctive properties, although they are not theoretically related.

To date, there are no reports of root shrinkage exceeding the original measured root length, which indicates a negative plastic deformation in halophyte roots (Figure 4). This phenomenon might be due to the high cellulose content of the cell walls in this species (Figure 5). Crystalline cellulose in kraft cooked Norway spruce showed that cellulose chains expand in a direction that is perpendicular to the cellulose chains after a tensile force is applied [38]. This response of the cellulose chains to tensile force may have caused the vertical shrinkage observed in the *S. salsa* roots. The high viscosity may also have contributed to the shrinkage in the 0 mM NaCl treatment (Figure 2). Possible factors contributing to this auxeticity might be the effect of microfibrillar orientation and shearing interactions between neighbor cells and cell wall layers [38,39]. A high amount of cellulose may provide tensile strength and crosslink sites that increase the cell wall stiffness and ensure the cell structure of *S. salsa* under salinity stress. This cell wall property may also limit the elongation of the roots in comparison to S. *oleracea* (Figure 1). The correspondence between viscoelastic extension, E_0_, and root length (Figure 3 and Figure 4) showed that cell wall elasticity was an important factor in determining cell elongation and, thus, root length in both halophyte and glycophyte species. Hattori et al. [24] also reported that mechanical cell wall properties determine root growth under drought conditions. Compared to the 0 mM NaCl treatment, less plastic shrinkage in the higher NaCl treatments indicated that salt treatment stiffened the cell walls in *S. salsa* (Figure 1, Figure 2 and Figure 4).

### Chemical Composition of the Root Cell Wall

The pectin content in the cell walls of *S. oleracea* roots was consistently affected by salinity (Figure 5). These results revealed the sensitivity of pectin to salinity in this glycophyte. Pectin is known to have many important functions in plant meristems, such as ion binding, ion homeostasis, pH adjustment, water retention, and electro-chemical balance [40]. The Ca^2+^–pectate and esterification pectin are also critical in controlling the chemical properties and the cell walls’ viscoelasticity by affecting cross-linkages to cell wall polymers [8,41,42]. These functions may have been inhibited under saline conditions in *S. oleracea* and resulted in reduced root growth. The reduced pectin content in *S. oleracea* after saline treatment may have reduced cell wall extensibility, which subsequently reduced root growth (Figure 1, Figure 2, Figure 3, Figure 4 and Figure 5, Table 3).

Cell wall pectin has been found to play a key role in cation binding [43] because the galacturonic acids in pectin provide cation binding sites [44]. The uronic acid content in the pectin fraction of cell walls was significantly correlated with pectin content in *S. oleracea* in response to salinity (Figure 5; Table 3). Furthermore, the proportion of uronic acid content in pectin increased as salinity increased in both species (Figure 5). This finding indicated that uronic acid is an important functional constituent of pectin that plays a significant role in plant tolerance and adaptation to adverse stress conditions. The increased proportion of uronic acid in pectin after NaCl treatment suggested that NaCl enhanced uronic acid synthesis, which may be related to its cation binding ability. Additionally, the lack of glucuronic acid could increase cell wall thickness [45], and it may also relate to plant salt tolerance [46]. Höfte et al. [42] and Le Gall et al. [47] reported that the degree of pectin methyl esterification was changed under salt stress, likely affecting the structural integrity and the mechanical properties of the cell wall. The absolute contents of pectin and uronic acid were changed under salinity treatment. Whether these changes affect the degree of pectin methyl esterification is not clear. The interactions of galacturonic acid, pectin methyl esterification and root growth under salinity, then, need to be further investigated. Changes in cell wall extensibility may occur through the synthesis of new cell wall material or changes in cell wall polysaccharides [26]. We found that NaCl inhibited pectin synthesis while enhancing cellulose synthesis in *S. oleracea*. Thus, pectin content decreased while cellulose content increased, especially at 200 and 300 mM NaCl (Figure 5 and Figure 6). These findings may explain the decrease in cell wall extensibility observed in the plants grown in the high saline treatment conditions (Figure 2). Therefore, NaCl treatment could affect root growth in glycophytes through changes in cell wall synthesis (Figure 5 and Figure 6, Table 1). In contrast, the root growth in halophytes was unrelated to cell wall pectin synthesis.

The primary salinity tolerance mechanism involved in *S. salsa* is Na^+^ accumulation and compartmentation in the vacuole [31]. Therefore, the weak correlations between pectin content or uronic acid content (in the pectin fraction) and root growth in *S. salsa* showed that pectin might not be related to salt tolerance in this species (Table 2). The significant increase in hemicellulose I, hemicellulose II, and cellulose content with increased concentrations from 100 mM to 300 mM NaCl may have reduced cell wall extensibility, which inhibited root growth in *S. salsa* (Figure 1, Figure 2, Figure 3 and Figure 5). Indeed, the fact that both species showed increased cellulose content with increased salt treatments from 100 mM to 300 mM NaCl suggested that cellulose is an important structural constituent, which is affected by salinity in both halophytes and glycophytes (Figure 5 and Figure 6). Given that increased cellulose synthesis can contribute to the preservation of cell wall integrity and rigidity [15], the biosynthesis of cellulose in both species probably contributed to the maintenance of cell morphology and assisted in salt stress resistance. In *S. salsa*, changes in pectin, hemicellulose, and cellulose content had the same pattern in all NaCl treatments; however, only pectin showed a difference at 300 mM NaCl (Figure 5). *S. salsa* is a halophyte that accumulates biomass even in high saline conditions [31]. The effects of salt stress on cell wall synthesis and damage to pectin- and cellulose-related components can alter salt tolerance [48,49]. We speculate that, due to its specific salt-tolerance mechanisms and because it contains almost twice as much hemicellulose and cellulose as *S. oleracea*, *S. salsa* has a more stable cell wall architecture in salt stress conditions (Table 1; Figure 5). A highly saline condition disrupts cross-linking between pectin and other cell wall fractions [7], which affects the stability of the cell wall and significantly inhibits root growth. A stable cell wall architecture may contribute to the overall salt tolerance of *S. salsa*.

Our study showed that, for two contrasting members of the Amaranthaceae, the halophyte *S. salsa* and glycophyte *S. oleracea*, the effects of salinity stress on root growth are closely related to the mechanical properties and chemical composition of the cell wall. Salinity affects root growth through the processes of cell wall loosening and synthesis. Cellulose may provide mechanical strength, but it limits root elongation under saline conditions. In halophytes, the high content of the cell wall and the proportion of cellulose in the cell wall may be a salt tolerance mechanism that protects the cell structure’s stability under salt stress. Pectin played important roles in both the glycophyte and halophyte, but the effect was more pronounced in the glycophyte, possibly because of their different salt tolerance mechanisms. The glycophyte relies on salt exclusion, whereas the halophyte relies on salt absorption and compartmentation in vacuoles. The function of the root cell wall in root growth was more prominent in the glycophyte than in the halophyte under saline conditions.

## Figures and Tables

**Figure 1 plants-11-00900-f001:**
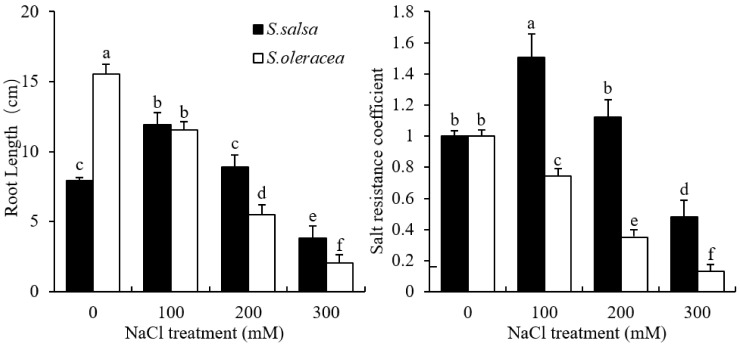
Final root growth and salt resistance coefficient of *S. salsa* and *S. oleracea* in 0, 100, 200, and 300 mM NaCl treatments. Data are mean + S.D. (*n* = 5). Salt resistance coefficient was calculated as: root length of NaCl treatments/root length of 0 mM NaCl. Different letters indicate significant differences (*p* < 0.05).

**Figure 2 plants-11-00900-f002:**
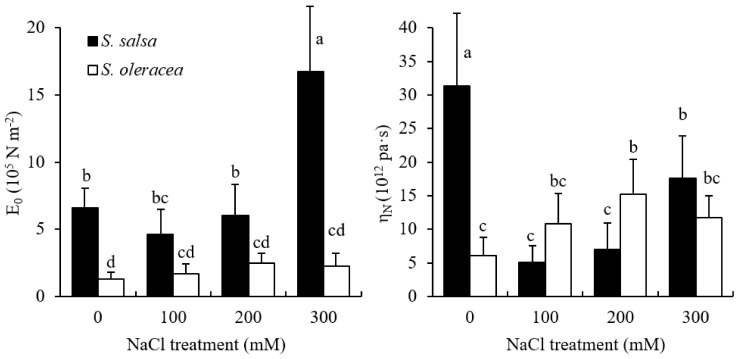
Elastic moduli (E_0_) and viscosity coefficient (η_N_) of the root cell wall in the elongation zone in *S. salsa* and *S. oleracea* in 0, 100, 200, and 300 mM NaCl treatments. Data are mean + S.D. (*n* = 12–15). Different letters indicate significant differences (*p* < 0.05).

**Figure 3 plants-11-00900-f003:**
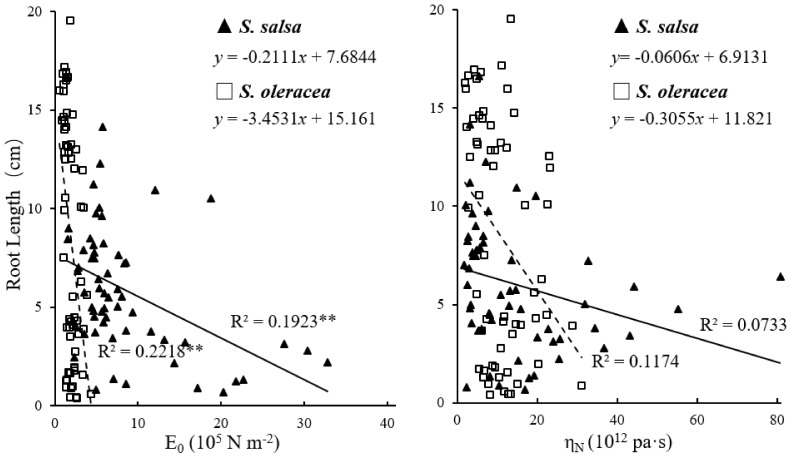
Correlation of root length with E_0_ (elastic modulus; left panel) and η_N_ (viscosity coefficient; right panel) in *S. salsa* (—) and *S. oleracea* (- - -) under 0, 100, 200, and 300 mM NaCl treatments. ** significant at *p* < 0.01 (*n* = 56–60).

**Figure 4 plants-11-00900-f004:**
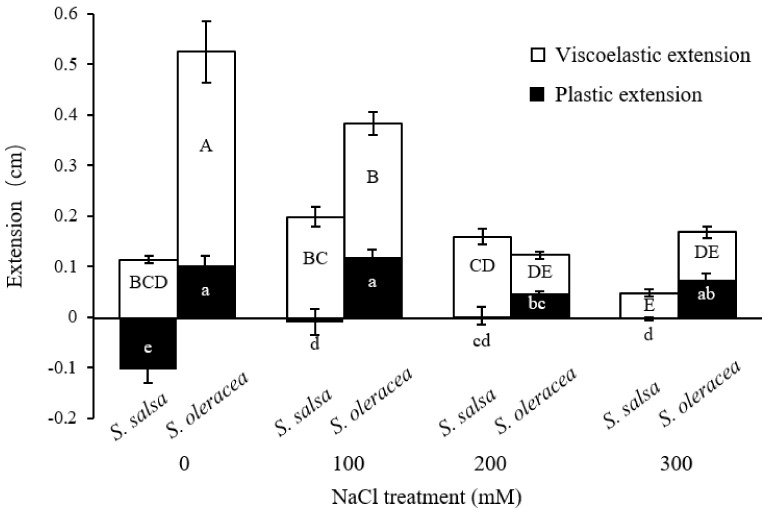
Viscoelastic extension and plastic extension in *S. salsa* and *S. oleracea* root grown under 0, 100, 200, and 300 mM NaCl treatments. Data are mean ± S.D. (*n* = 12–15). Different uppercase (viscoelastic extension) and lowercase (plastic extension) letters indicate significant differences (*p* < 0.01).

**Figure 5 plants-11-00900-f005:**
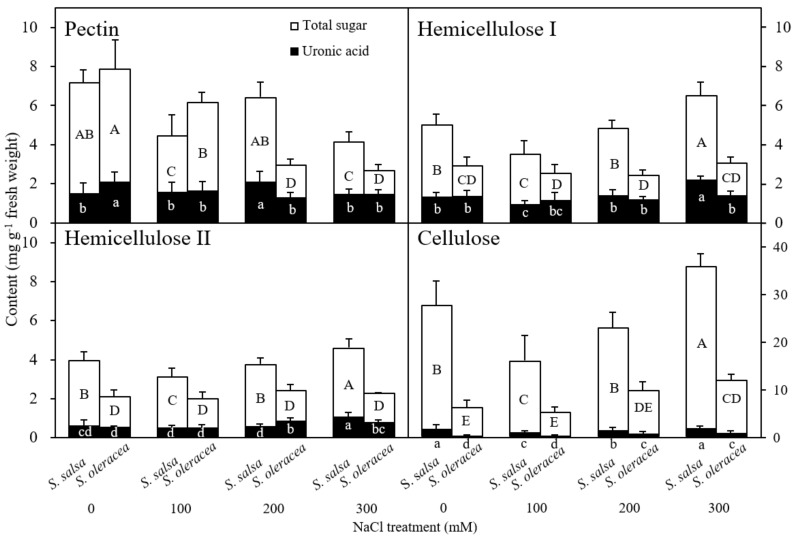
Total uronic acid-containing sugar and uronic acid content in the pectin, hemicellulose I, hemicellulose II, and cellulose cell wall fractions in the apical 0–5 mm region in *S. salsa* and *S. oleracea* roots grown under 0, 100, 200, and 300 mM NaCl treatments. Data are mean + S.D. (*n* = 4). Different uppercase (total sugar) and lowercase (uronic acid) letters indicate significant differences (*p* < 0.05).

**Figure 6 plants-11-00900-f006:**
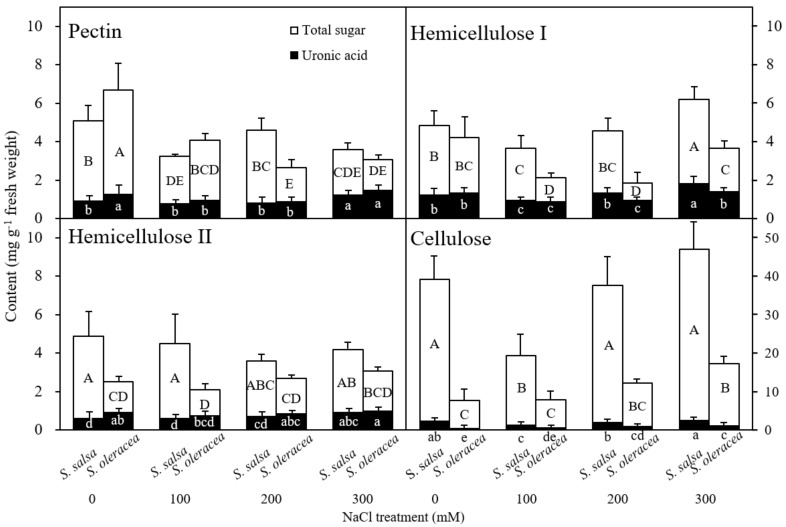
Total uronic-acid-containing sugar and uronic acid content in the pectin, hemicellulose I, hemicellulose II, and cellulose cell wall fractions in the apical 5–10 mm region in *S. salsa* and *S. oleracea* roots grown under 0, 100, 200, and 300 mM NaCl treatments. Data are mean + S.D. (*n* = 4). Different uppercase (total sugar) and lowercase (uronic acid) letters indicate significant differences (*p* < 0.05).

**Table 1 plants-11-00900-t001:** Total sugar content in the cell wall of the 0–5 mm and 5–10 mm root segments in *S. salsa* and *S. oleracea* young seedlings grown under different concentrations of NaCl. Data indicate mean ± S.D. (*n* = 4).

Species	Root Region	Sugar Content (mg g^−1^ FW)			
		NaCl Treatment (mM)			
		0	100	200	300
*S. salsa*	0–5 mm	43.5 ± 6.4 ab	26.8 ± 7.3 c	37.7 ± 3.7 bc	50.7 ± 3.0 a
*S. oleracea*		18.8 ± 2.3 a	15.7 ± 1.1 b	17.4 ± 1.7 a	19.7 ± 1.1 a
*S. salsa*	5–10 mm	53.9 ± 7.5 ab	30.6 ± 7.9 b	52.9 ± 13.7 ab	61.0 ± 10.3 a
*S. oleracea*		21.1 ± 4.5 ab	16.1 ± 2.4 b	19.3 ± 1.4 b	27.0 ± 2.0 a

Different lowercase letters indicate significant difference at *p* ≤ 0.01, *n* = 4. FW, fresh weight.

**Table 2 plants-11-00900-t002:** Cross-correlation coefficients of final root length (RL); total sugar content of pectin, hemicellulose I (HC-I), hemicellulose II (HC-II), and cellulose; uronic acid (UA) content of pectin, HC-I, HC-II, and cellulose in the 0–5 mm region of apical root cap in *S. salsa*.

	RL	Pectin	HC-I	HC-II	Cellulose	Pectin (UA)	HC-I (UA)	HC-II (UA)
Pectin	0.121							
HC-I	−0.868 **	−0.008						
HC-II	−0.814 **	0.005	0.915 **					
Cellulose	−0.882 **	0.04	0.821 **	0.827 **				
Pectin (UA)	0.148	0.287	−0.081	−0.005	−0.003			
HC I (UA)	−0.929 **	−0.268	0.900 **	0.825 **	0.849 **	−0.136		
HC II (UA)	−0.761 **	−0.329	0.766 **	0.739 **	0.811 **	0.048	0.821 **	
Cellulose (UA)	−0.724 **	0.268	0.786 **	0.854 **	0.875 **	−0.016	0.657 **	0.647 **

* Correlation is significant at the 0.05 level (two-tailed); ** Correlation is significant at the 0.01 level (two-tailed). (*N* = 14).

**Table 3 plants-11-00900-t003:** Cross-correlation coefficients of final root length (RL); total sugar content of pectin, hemicellulose I (HC-I), hemicellulose II (HC-II), and cellulose; uronic acid (UA) content of pectin, HC-I, HC-II, and cellulose in the 0–5 mm region of apical root cap in *S. oleracea*.

	RL	Pectin	HC-I	HC-II	Cellulose	Pectin (UA)	HC-I (UA)	HC-II (UA)
Pectin	0.939 **							
HC-I	−0.055	0.113						
HC-II	−0.417	−0.497	0.276					
Cellulose	−0.826 **	−0.784 **	0.118	0.513 *				
Pectin (UA)	0.718 **	0.851 **	0.459	−0.249	−0.546 *			
HC I (UA)	−0.033	−0.068	0.666 **	0.535 *	0.274	0.214		
HC II (UA)	−0.787 **	−0.845 **	0.117	0.811 **	0.790 **	−0.573 *	0.339	
Cellulose (UA)	−0.884 **	−0.862 **	−0.006	0.576 *	0.958 **	−0.650 **	0.196	0.857 **

* Correlation is significant at the 0.05 level (two-tailed); ** Correlation is significant at the 0.01 level (two-tailed). (*N* = 14).

## Data Availability

Data sharing is not applicable to this article.

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
