# Peer review of "Cell Wall Components and Extensibility Regulate Root Growth in *Suaeda salsa* and *Spinacia oleracea* under Salinity"

_plants, 2022, doi:10.3390/plants11070900_

Round 1
Reviewer 1 Report
The manuscript “Cell wall components and extensibility regulate root growth in Suaeda salsa and Spinacia oleracea under salinity”, written by Jia Liu, Ping An, Yang Shao, Xiaohui Feng, Asana Matsuura, Irshad Muhammad, Yuanrun Zheng, comprises an article of high value for the field of plant sciences.
The biological mechanisms of root growth correlated with the composition and mechanical capacity of the cell wall at increasing concentrations of NaCl are well developed in this work, in a strict halophyte (Suaeda salsa) and a glycophyte (Spinacia oleraceae).
Introduction, materials and methods, results and discussion are concise, very clear and well written and developed sections. The literature cited is adequate in the framework presented. The graphs and tables presented are accurate and with axes in the appropriate units.
However, I include some light considerations to deal with in the work, labeled under the “minor revisions” statement.
Line 101: add authorship in Spinacia oleracae L. the first time it is cited.
Line 107: remove L. in Suaeda salsa (authorship cite only the first time).
Line 113: plural vacuoles better than vacuole?
Other general comments:
- We suggest incorporating a figure with stylized photographs of root growth in some of the experiments. This can trigger a more visual and aesthetic work.
- Spinacia and Suaeda, are two genera that are currently accepted by the majority under the Amaranthaceae family, please adjust throughout the document, and comment, where appropriate, that they were formerly grouped in the Chenopodiaceae family.
Reviewer 2 Report
The manuscript “Cell wall components and extensibility regulate root growth in Suaeda salsa and Spinacia oleracea under salinity” by Jia Liu et al. is an interesting piece of work addressing the role of cell wall components and functionality under salt stress. Analysis of contrasting responses of an halophyte and a glycophyte revealed important insights in the mechanisms of adaptation/tolerance to salinity which may lead to new targets for genetic improvement of salt stress tolerance in agricultural crops. A few minor criticisms:
- Figure 2 is missing the statistics
- Table 2 is not easy to read – format problems
- The manuscript would greatly benefit of rearranging Results and Discussion into two separate sections. This would allow authors to propose a model for coordination of responses between root biochemical/anatomical modifications and the most known shoot responses to high salinity of the root zone.
Reviewer 3 Report
The authors used mechanical tests and performed chemical analysis of the cell wall to relate them to growth response of halophyte and glycophyte plants subjected to salinity stress. The authors showed increase of cellulose content and pectin content in the one of species under the stress.
Concerning physical measurement, the authors postulated a relationship between measured parameters and growth inhibition.
The authors tackled difficult task to related biomechanics to growth that require both knowledge of biomechanical basis of applied tests and understanding the growth mechanism.
The use of mechanical test and parameters in this context may lead to various confusions, which were, unfortunately, not avoided in this manuscript.
Abstract
Line 32…. Meanwhile, S. salsa root cell walls exhibited high mechanical strength to resist deformation 34 under 0 and 300 mM NaCl treatment.
- “high is aways relative” – we can rather say “higher than/compared to…”
- Mechanical strength was not measured by the authors- Please check definition of “mechanical strength”.
- Resistance to deformation is defined by stiffness, and not by strength. The strength is a measure of sample “resistance to damage” but not to deformation.
Lines 38-44.
Our results suggest that 1) cell wall pectin plays important roles in cell wall extensibility in both species under salinity and that the salt tolerance of glycophyte S. oleracea is directly affected by the pectin content
- No argument for this strong statement- on the base of the data - the increase or lack of changes of pectin content was observed; cellulose content was increased in both species
- So it is just an association of measured features but not a proof for their effect. Moreover, it is well documented that not amount of compounds but rather the interlinks between compounds are important for wall mechanics. However, the latter were not measured here.
The growth rate modulation is affected rather by extent of pectin methylesterification and not by pectin content itself. The authors did not mention about this factor in the manuscript.
Methods
Lack of data on the amount of applied force in the creep test.
Line 144 ….. “The difference in length between the maximum extension 300 s and the final length after 600 s was reported as a reversible extension (i.e., elastic extension)…
- Reversible extension in not in this case pure elastic extension ( rather viscoelastic extension). Pure elastic extension recovery is immediate, if it takes longer period we define as viscoelastic deformation.
Line 154 ….”Approximately 96-120 segments from six filter paper sheets were collected as one replicate, and four 156 replicates were measured from each treatment”. – not clear meaning for me
Statistics
The authors applied the Duncan MRT test for pairwise comparison of means.
“Duncan's test has been criticised as being too liberal by many statisticians… “
“ Duncan’s tests is based on the Newman–Keuls procedure, which does not protect the familywise error rate “
Why did the authors not use the Tukey HSD test?
The authors used SE to characterise plotted data. But for data presented in Table1 the SD is used. Why?
The Standard Error (SE) values depend on sample size and more valuable descriptive statistics is provided by the standard deviation (SD) - much less dependent on the sample size. What was the reason to use SE instead SD for some data.
Results
Line 211 …. “The difference from E0 was that the value of 10 0 mM NaCl treatment in ηN rose dramatically in S. salsa” - sounds unclear
Line 218 ………. “Salt stress causes significant cell wall stiffness”
Stiffness is a mechanical parameter and not a description of a processes.
We can say “cause …. stiffening” but not “ cause .. stiffness”.
Line 237… “It also showed that extensibility and viscosity are distinctive properties, although they are closely related”.
– Indeed quite different by definition but they are not related on the basis of the theory – they may increase parallelly for some material, like here.
Line240 ….”, negative plastic extension values were measured” – very strange and interesting behaviour. Contraction possibly due to the loss of water?
Plastic extension defined as an irreversible extension typically cannot be negative.
Concerning extensibility – the wall feature which is related to growth rate in literature
The authors assumed the E0, i.e. the elastic modulus/stiffness is a measure of extensibility, if I well understand.
However, it was emphasised many times in literature ( see papers by Cosgrove) that cell wall extensibility is not equivalent to Young’s modulus or stiffness in extension.
Stiffness is measure of resistance to deformation (reversible) but growth is related to irreversible behaviour. Thus significant confusion in use parameters and their interpretation.
Line 551 incorrect citation:
Höfte, H., Peaucelle, A., & Braybrook, S. (2012). Cell wall mechanics and growth control in plants: the role of pectins revisited. Frontiers in Plant Science, 3, 121.
Summarising, the authors showed some association between measured parameters and growth response under salinity stress.
In my opinion, extensibility ( in the reversible meaning) was not measured here. The E0 characterises stiffness i.e. resistance to deformation but not to growth ( irreversible phenomenon).
The manuscript could be an interesting contribution showing some associations between measured features and growth response to stress but I suggest to perform it with much higher cautions avoiding misuse of term “extensibility”.
